# Shifts in Diabetes Health Literacy Policy and Practice in Australia—Promoting Organisational Health Literacy

**DOI:** 10.3390/ijerph20105778

**Published:** 2023-05-10

**Authors:** Giuliana O. Murfet, Shanshan Lin, Jan C. Ridd, Gunhild H. Cremer, Susan Davidson, Danielle M. Muscat

**Affiliations:** 1School of Public Health, University of Technology Sydney, Ultimo, NSW 2007, Australia; shanshan.lin@uts.edu.au; 2Diabetes Centre, Tasmanian Health Service, Burnie, TAS 7320, Australia; 3Diabetes Australia, Turner, ACT 2612, Australia; jridd@diabetesaustralia.com.au (J.C.R.); gcremer@diabetesaustralia.com.au (G.H.C.); 4Australian Diabetes Educators Association, Chifley, ACT 2606, Australia; susan.davidson@adea.com.au; 5Sydney Health Literacy Laboratory, School of Public Health, Faculty of Medicine and Health, The University of Sydney, Sydney, NSW 2006, Australia; danielle.muscat@sydney.edu.au

**Keywords:** organisational health literacy, diabetes, health information, diabetes Australia, national diabetes services scheme, evaluation

## Abstract

Improving organisational health literacy ensures people can navigate, understand and use essential health information and services. However, systematic reviews have identified limited evidence for practical approaches to implementing such organisational change, particularly at a national level. This study aimed to (a) investigate the approach taken by an Australian national diabetes organisation—Diabetes Australia, as the administrator of the National Diabetes Services Scheme (NDSS)—to improve organisational health literacy over a 15-year-period and (b) examine the impact of organisational changes on the health literacy demands of health information. We performed an environmental scan, examining the websites of the NDSS, Diabetes Australia and the Australian government for reports and position statements describing organisational health literacy policies and practices between 2006 and 2021. The Patient Education Materials Assessment Tool (PEMAT) was applied to consecutively published NDSS diabetes self-care fact sheets (n = 20) to assess changes in the health literacy demands (understandability and actionability) of these fact sheets over the same period. We identified nine policies resulting in 24 health literacy practice changes or projects between 2006 and 2021, applied using a streamlined incremental approach and group reflexivity. The incremental approach focused on (1) increasing audience reach, (2) consistency and branding, (3) person-centred language and (4) the understandability and actionability of health information. The PEMAT scores of fact sheets improved between 2006 and 2021 for understandability (53% to 79%) and actionability (43% to 82%). Diabetes Australia’s information development process leveraging national policies, employing an incremental approach and group reflexivity has improved the health literacy demands of diabetes information and serves as a template for other organisations seeking to improve their organisational health literacy.

## 1. Introduction

Australian and international health policies increasingly recognise the need to address health literacy [1], reflecting an increased awareness of health literacy as a critical determinant of quality and safe clinical care and the high prevalence of low health literacy [2]. While international estimates vary, systematic reviews and large-scale surveys have identified that 47% to 55% of adults globally have low health literacy [3,4], costing national governments at least US$117 billion annually [5].

In Australia, only two in five adults (40%) have “adequate” health literacy, i.e., the level of individual health literacy required to meet the demands of everyday life and understand and follow health messages [6]. Further, significant social disparities exist by age, language spoken at home and rurality. The most recent national survey of health literacy, for example, identified differences in health literacy domains by age and language spoken, with more people who spoke English at home strongly agreeing they felt understood and supported by healthcare providers (33%) than those who did not (20%) [6]. Moreover, only 36% of Australian adults had adequate health literacy in rural areas compared to 42% in urban areas.

Studies examining a country’s functional health literacy have found that those countries with inadequate health literacy had a significantly higher incidence of chronic conditions than those with adequate health literacy, particularly diabetes [7,8,9,10]. Simultaneously, health literacy impacts the effectiveness of chronic condition management, such as diabetes [9]. Inadequate health literacy is associated with worse glycaemic control and higher rates of diabetes-related micro- and macrovascular complications, particularly retinopathy, and mortality [11,12,13]. Specifically, people with diabetes and low health literacy have greater difficulty understanding their condition and participate less in self-care activities, increasing the burden of diabetes [14]. Improving health literacy can, therefore, be seen as one tool in enabling effective partnerships in health and increasing engagement in self-care to reduce the personal and economic burdens of diabetes in Australia [15,16,17].

*Organisational health literacy* is an organisational-wide effort to transform an organisation’s delivery of care and services to make it easier for people to navigate, understand and use information and services to care for their health [18]. Seminal work by the Institute of Medicine’s Roundtable on Health Literacy identified ten attributes of health literate organisations [19]. Among others, these include leadership that promotes health literacy integral to its mission, structure and operations; co-design; implementation and evaluation of health information and services; culture of innovation; and the distribution of print, audio–visual and social media content that is easy to understand and act on [19].

Since then, several reviews have focused on effective strategies for creating health literate organisations, providing insight into implementation barriers and enablers [18,19,20,21,22]. Based on their review, Charoghchian Khorasani and colleagues concluded that shifting to a health literate organisation requires radical, concurrent and multiple changes, because integration is complex and health literacy is rarely integrated into healthcare organisations’ vision and strategic planning [20]. Simultaneously, Lloyd et al. recognised that current approaches are often inadequate for producing the changes needed to improve organisational health literacy [22]. In Australia, developing a national health literacy strategy to guide health service improvements is one priority under the new “National Preventive Health Strategy”. Currently, in a consultation phase, its objective is to provide an evidence-based health literacy environment where health information is person-centred, accessible and culturally and linguistically appropriate and to improve the health literacy skills of all Australians [8,23].

The aim of this study was to investigate and describe the approach that Diabetes Australia took to improve organisational health literacy and to examine the impact of organisational changes on the understandability and actionability of health information over 15 years. By describing the organisational changes enacted by Diabetes Australia, this manuscript aims to provide a practical example that can serve as a template for other organisations seeking to improve their organisational health literacy and guide policy.

## 2. Method

### 2.1. Context and Setting

Australia is a multicultural country with large geographical areas of remoteness. The prevalence of diabetes increases in populations living farther from urban areas and is three times more common in Aboriginal and Torres Strait Islander people [24,25]. Over two-thirds of Australians live outside urban metropolitan areas, whereas approximately 90% of the medical diabetes specialist workforce works in urban areas [26,27].

Diabetes Australia is the national organisation for the 1.5 million Australians diagnosed with diabetes and those at risk in Australia [24,28]. Diabetes Australia is committed to reducing the impact of all types of diabetes and works in partnership with people with diabetes, health professionals and researchers [28].

On behalf of the Australian Government, Diabetes Australia administers the National Diabetes Services Scheme (NDSS) through multi-year NDSS Agreements [28,29]. The NDSS aims to enhance the capacity of people with diabetes to understand and self-manage their condition through access to information, education, support services and subsidised diabetes products to minimise the impact of diabetes on their lives and the community [29]. To support this process, the NDSS maintains a national register of those diagnosed with diabetes who apply for subsidy access, and the diabetes workforce promotes registration throughout Australia [29]. An NDSS Agreement is a contract between the Australian government (represented by the Department of Health) and the NDSS administrator. The NDSS Agreement stipulates the expected outcomes of funding provided and project management and evaluation measures.

### 2.2. Study Design

This study used a nonexperimental and descriptive design with an environmental scan and Patient Education Materials Assessment Tool (PEMAT) analysis to examine the approach that Diabetes Australia took to improve organisational health literacy and its impact. No ethical approval was required. A diagram of the entire study design and timing is provided in the Appendix A (see Appendix A).

### 2.3. Environmental Scan Method

An environmental scan of Diabetes Australia and the NDSS, a method of seeking, gathering, interpreting and analysing information from the internal and external environments of an organisation [30,31], was undertaken to understand and describe the changes in organisational health literacy over three NDSS Agreement periods between 2006 and 2021. A preliminary literature review (conducted in September 2021) on CINAHL and Medline using the search terms “diabetes” and “organisational OR organizational” and “health literacy” did not identify any peer-reviewed literature describing diabetes-specific organisational health literacy.

#### 2.3.1. Grey Literature Search Strategy

We examined the websites of the NDSS, Diabetes Australia and the Australian government for reports, position statements, minutes and documents developed between 2006 and 2021. Documents were selected based on their content’s relevance to the inquiry; any information using the terms “diabetes” and “health literacy” was examined to identify changes in organisational health literacy policy, including whether health any literacy practice changes occurred within the organisation because of the policy or report.

#### 2.3.2. Data Extraction, Synthesis and Analysis Procedure

After the initial searches were completed by two researchers (G.O.M. and G.H.C.), the identified documents and policies were listed in an Excel document (G.O.M.), with a summary describing the actions delivered or advice listed to improve health literacy, and this was cross-checked by a policy expert (G.H.C.) (see Appendix A). During the data analysis period, the summaries were shared with two researchers (S. L. and G.O.M.) for content analysis using an inductive approach and emergent coding to identify patterns or activities and categorise the information into themes [32]. Next, the themes were reviewed during the key informant consultation, discussed at point 2.3.3.; subsequently, a third researcher (D.M.M.) then supported finalising major themes.

#### 2.3.3. Key Informant Consultation

In alignment with the key informant consultation process [33,34,35], the preliminary findings were presented to the larger research team comprised of researchers and policymakers with expertise in diabetes, health literacy and diabetes policy (see Appendix A). We consulted executives from Diabetes Australia and Australian Diabetes Educators Association (ADEA), those involved in developing or implementing the policies over the 15 years (J.C.R., G.H.C. and S. D.) to confirm the accuracy of the interpretation of the chronologies and summaries generated (i.e., the intent of the document and audience) to improve the coding [36] and enhance themes generated (see Appendix A).

In addition, to facilitate the inclusion of nonexecutive experts who offered advice to Diabetes Australia, our key informant consultation included health professionals providing health literacy advice (a dietitian and a nurse) from the Medical Educational and Scientific Advisory Council (MESAC). The key informants were asked clarifying questions, including whether the summaries accurately reflected what was happening and/or the purpose of the tool/activity and how these might be used or discussed during meetings.

This consultation improved the understanding of programmes, resources and processes and supported the identification of superseded documents or additional relevant material, e.g., updated NDSS health literacy guided checklists or branding guides. Last, enhance identification of major themes.

### 2.4. Method for the Assessment of Understandability and Actionability Using PEMAT

To assess the impact of the implemented changes in organisational health literacy, we examined changes in the understandability and actionability of diabetes self-care fact sheets, which are one form of health information. A search was conducted by two researchers (G.H.C. and G.O.M.) in September to October 2021 to obtain accessible fact sheets (in the English language) published by Diabetes Australia during each NDSS Agreement period (2006, 2010, 2016 and 2021).

Nine fact sheets, known as diabetes self-management education (DSME) information, were available at all time points. Then, five fact sheets were randomly selected using an online random number generator “https://numbergenerator.org/ (accessed on 10 October 2021)”. We excluded non-English written information in pictorial or graphic form or videos. An earlier study examining the readability, understandability and actionability of similar health information categorised fact sheets into three main content categories: general, medical and lifestyle information [37]; these were applied to this study.

Shoemaker and colleagues developed and validated the PEMAT for printable materials to assess understandability using 17 items and actionability using 7 items [38]. A 70% or higher score indicates reasonable understandability and actionability [39]. The consistent and reliable results found among diverse assessors suggest that healthcare professionals and laypersons can use the PEMAT [38,39]. Further, PEMAT has been used to assess diabetes health information, including Australian fact sheets [37,40,41].

Two researchers (S. L. and R. K.) independently performed the evaluation; both were experienced with PEMAT and trained according to the User’s Guide. Each item was scored on a binary scale (agree (1) or disagree (0)), except for seven items, which included a “not applicable” option. The results are expressed as the percentage of items coded “agree”, and a higher percentage suggests that the fact sheet is more likely to be accessible and that it is easier for a person to act on the information presented. A third researcher (D. M.M.) was consulted when a discrepancy could not be reconciled (see Appendix A).

## 3. Results

### 3.1. Policy and Practice Changes to Improve Organisation Health Literacy

Diabetes Australia, through the NDSS, implemented 24 health literacy-related policy and practice changes since 2006 (see Figure 1). These broadly align with nine health literacy initiatives, including national surveys and the National Statement on Health Literacy [42,43,44], and have been actioned through consecutive NDSS Agreements between the Australian Department of Health and Diabetes Australia (see Figure 2).

In the first instance, organisational health literacy responses often centred on written information, including four foci: (1) expanding audience reach; (2) consistency and branding; (3) person-centred language; and (4) reducing health literacy demands (i.e., readability, understanding and actionability) of written DSME information (see Appendix A for definitions and Appendix A for narratives from key informants and OHL response to each focus). Moreover, an incremental process was used, supported by a health literacy guided form and group reflexivity. The following section elaborates on the key findings for each agreement period.

#### 3.1.1. The 2006 to 2011 National Diabetes Services Scheme Agreement Period

The findings from the “2006 Adult Literacy and Life Skills Survey” were released in 2008, spotlighting the prevalence of low health literacy in Australia, particularly for migrants from non-English-speaking countries [42]. In response, Diabetes Australia formed the “Medical, Education and Scientific Council” (MESC), an advisory body that included representation from diverse healthcare professionals (see Figure 2 and Appendix A) [45]. The purpose of MESC was to provide an independent appraisal of all DSME information (fact sheets and audio–visual resources) developed and delivered by Diabetes Australia, compared against evidence-based literature, and to advise Diabetes Australia [45]. Diabetes Australia also had the ten most commonly accessed DSME fact sheets translated into ten different languages (see Figure 3 and Appendix A) [46].

Concurrently, in 2008, most diabetes associations (consumer advocacy organisations from Australian states and territories), the Australian Diabetes Educators Association (peer organisation for Credentialled Diabetes Educators—nursing and allied health professionals) and the Australian Diabetes Society (peer organisation for medical diabetes experts) became subcontracted “Agents” (recipients of funding under NDSS Agreements) to Diabetes Australia. Bringing Agents together through Diabetes Australia created greater opportunities for national consistency in developing DSME information and services for people living with diabetes. Until 2007, state and territory diabetes associations independently developed all DSME information (e.g., fact sheets) and programmes, leading to duplication [47].

#### 3.1.2. The 2011 to 2016 National Diabetes Services Scheme Agreement Period

The 2011 to 2016 NDSS Agreement stipulated that Diabetes Australia establish a “Medical Education and Scientific Advisory Council” (MESAC) to provide an independent appraisal of NDSS-funded DSME information and to advise the NDSS [48]. The MESAC’s scope had moved beyond the role of MESC to include the evaluation of the educational soundness of information, including the appraisal of health literacy demands to ensure tailoring to people’s needs, knowledge and skills. The MESAC met with the NDSS regularly to enable the inquiry and exploration of options and an array of opinions that fed into updates to the NDSS health literacy guided checklist, later used as a tool by authors submitting diabetes health information to support health literacy (see Appendix A).

Simultaneously, Diabetes Australia launched the “A New Language for Diabetes Position Statement” describing language that engages, motivates and is preferred by people living with diabetes (see Figure 1, Figure 2 and Appendix A) [49]. Further, Diabetes Australia appointed NDSS Priority Area Leaders, champions of inquiring and advocating for vulnerable populations living with diabetes, including older and younger, culturally and linguistic diverse, and Aboriginal and Torres Strait Islander populations, as well as people living with psychosocial issues and diabetes in pregnancy. In addition, Diabetes Australia launched an annual NDSS registrant survey to identify the number of fact sheets being accessed and the awareness of these fact sheets among registrants (see Figure 3).

The Australian Commission on Safety and Quality in Healthcare released the “National Statement on Health Literacy” in 2014, acknowledging health literacy as a safety and quality issue [44]. Resultantly, in 2014, the Australian Diabetes Educators Association and NDSS released a “Health Literacy Position Statement” and toolkit to build literacy skills [50]. The position statement recommended health literacy practices for credentialled diabetes educators delivering DSME, including (1) providing evidence-based information, (2) sharing decision making, (3) promoting health literate services and (4) developing cognitively and culturally appropriate co-designed education programmes in supportive health services environments [50].

The toolkit included practical resources to evaluate and improve the quality of clinical practices. It included an anonymous consumer survey to capture viewpoints on credentialled diabetes educators’ performance in delivering care compared to person-centred care principles, as well as a consumer interview designed to capture detailed feedback for planning health literacy improvements to the service [50]. Simultaneously, Diabetes Australia launched the “NDSS Style Guide” in 2015, guiding authors of DSME information on consistency in branding, presentation and the use of preferred language in alignment with the Diabetes Australia language statement (see Figure 2 and Appendix A). Only when these principles were met, as shown on an NDSS “Health Literacy Guided Checklist Form” (see Appendix A), was the information accepted for MESAC appraisal.

#### 3.1.3. The 2016 to 2021 National Diabetes Services Scheme Agreement Period

In 2016, the Australian government developed the *Australian National Diabetes Strategy 2016–2020*, outlining the national response to diabetes, with Diabetes Australia represented on the National Diabetes Strategy Advisory Group [51]. A strong focus of the strategy was disseminating culturally appropriate information and programmes. In response, Diabetes Australia commissioned the “Translation Project” to extend the number of translated fact sheets to 26 languages (see Figure 3 and Appendix A).

Meanwhile, NDSS Priority Area Leaders, champions for vulnerable populations, advocated expanding the modality of information delivery to more digitalised formats, including podcasts, videos, animations and interactive presentations and platforms [45]. For example, this could be seen in the increase in digital online topic-specific resources to meet the needs of people accessing diabetes technology under the Australian Government Continuous Glucose Monitoring Initiative in 2017 [52].

The 2018 “National Health Survey” used a multidimensional health literacy assessment tool to assess nine domains, including peoples’ perceptions of feeling understood and supported by healthcare providers, navigating the healthcare system, and health information appraisal [43]. The NDSS “Brand and Style Guide and Preferred Language Checklist” updates, informed by people with diabetes, reflected these domains to improve health literate communication by health professionals (see Appendix A) [48]. Simultaneously, the NDSS webpage was enhanced to improve site navigation, e.g., consistent screen design and layout and theme adjustments were co-designed to improve keyboard use and visual experience [53].

An “NDSS Preferred Language Checklist” was developed to direct authors on essential health literacy principles, including using readability assessments, active voice and visual aids to reduce health literacy demands (see Appendix A). Subsequently, the NDSS “Health Literacy Guided Checklist Form” was aligned, creating efficiencies in the appraisal process and instructed authors to detail the target audience better, address readability scores, and the report authors’ expertise (see Appendix A) [54].

#### 3.1.4. The 2021 to 2024 National Diabetes Services Scheme Agreement Period

The Australian government recently released the *Australian National Diabetes Strategy 2021–2030* [55]. To align, the current NDSS Agreement 2021–2024 includes a large-scale independent evaluation of all NDSS products, programmes and services to demonstrate consistency and impact. An “NDSS National Consistency Policy” was launched and led to Diabetes Australia developing the National Consistency Taskforce. A key policy element is that people with diabetes receive support to improve their health literacy to make informed health-related decisions and actions [55].

### 3.2. Assessment of Understanding and Actionability

The changes in Diabetes Australia’s policy and practice over time are reflected in the increased number of fact sheets developed and the number of translated fact sheets available since 2006 (see Figure 3). A mean of 14 (range 2 to 23) fact sheets were translated for each language between 2006 and 2021 (see Appendix A). In addition, an annual “NDSS Registrant Survey” of over 2200 registrants between 2014 and 2021 showed that the awareness of fact sheets increased from 34% to 47–51% (see Figure 3).

The five fact sheets included in this analysis were categorised into three content categories in Table 1. The mean understandability and actionability scores improved between 2006 and 2021. Mean fact sheets scores were 53% and 43%, respectively, in 2006, which did not meet health literacy requirements in terms of understandability and actionability. By 2021, all fact sheets scored above 70% for understandability and actionability, except for the “Prediabetes” fact sheet under the general information category, which had the lowest mean actionability (67%) (see Table 1).

Across all fact sheets in the different agreement periods, one assessment item for understandability, “uses a visual aid whenever possible”, was adhered to less than 13.3% of the time, i.e., only the “Glycaemic index” in 2016 and 2021 met the requirement. The actionability item “use visual aid whenever possible to help take action” was adhered to 33.3% of the time, and only “Coeliac diseases” 2021, “Glycaemic index” 2016 and 2021, “Sick days for type 2 diabetes” 2016 and “Staying well with diabetes” 2021 met the requirement.

## 4. Discussion

The aim of this study was to investigate and describe the approach that Diabetes Australia took to improve organisational health literacy and to examine the impact of organisational changes on the understandability and actionability of health information over 15 years. Diabetes Australia, through the NDSS, implemented 24 health literacy-related policies and practice changes between 2006 and 2021, with concurrent improvements in the understandability and actionability of written fact sheets identified in our analysis. Increased accessibility to a wider range of health information in more languages and increased awareness of their existence were also identified.

### 4.1. Facilitators of Organisational Change to Promote Health Literacy

Reflecting on the process of organisational change, three key facilitators appeared to support organisational health literacy:National health literacy data and frameworks;Incremental change focusing on (a) audience reach, (b) consistency and branding, (c) person-centred language and (d) understandability and actionability;Group reflexivity.

#### 4.1.1. National Health Literacy Data and Frameworks

National health literacy data and frameworks often appeared to serve as the impetus for change. Diabetes Australia leveraged national policies and data related to health literacy to inform change, as well as drawing on data from the NDSS database, a registry of people living with diabetes [29]. In this way, change was facilitated in both a top-down and bottom-up manner, with policy leadership and the voice of people living with diabetes providing an enabling environment for health literacy change. Population registers are known to have strengths and weaknesses, including missing information [56]; however, the NDSS registrant database’s utility for capturing a more extensive array of an audience nationally when promoting health and health literacy information to improve outcomes is not established [13]. Yet, it could support addressing the health literacy and type 2 diabetes prevention goals of the *Australian National Diabetes Strategy 2021–2030* [55].

#### 4.1.2. Incremental Change

Diabetes Australia purposively implemented small, systematic steps to enable incremental changes. Although organisational health literacy is multidimensional, a focus on written information in the first instance has led to observable improvements in the distribution of print, audio–visual and social media content that is easy to understand and act on. This process was supported by a “Health Literacy Guided Checklist Form” that served as a tool for authors or programme developers to appraise their work against health literacy criteria. Changes were made inclusively—co-designed with consumers and experts focused on the four foci: audience reach, consistency and branding, person-centred language and understanding and actionability (see Appendix A).

*I.* 
*Audience Reach*


The DSME health information has become more accessible, including to people with culturally and linguistically diverse backgrounds, as evidenced by the increased awareness of the fact sheets and the substantial increase in the number of translated fact sheets found in our study. Subsequently, the improved accessibility of DSME information, led by Diabetes Australia and NDSS, could be a key factor in promoting a more health literate environment in Australia. The improvements in audience reach align with the Australian Commission on Safety and Quality in Health Care, which legislates health literacy for safe and quality healthcare [44], and the ten attributes of organisational health literacy [19]. Audience engagement and reach have been shown to build more enduring and mutually beneficial relationships with the public that can support building trust and strengthening ties with the targeted population [57]. A recent roundtable on health literacy reported that trust has a role to play in health literacy to achieve equity in healthcare; therefore, “trust” is an important element [58].

*II.* 
*Consistency and Branding*


The environmental scan identified that consistency and branding were essential to the “tools” Diabetes Australia used to build organisational health literacy. The literature suggests that consistent healthcare branding is an approach to help healthcare consumers identify with the institution or service they need despite their in-the-moment needs [59,60]. It builds trust and promotes the organisation’s role in maintaining the health and welfare of communities [59].

*III.* 
*Person-centred Language*


The focus on language to support health literacy is now accepted as good practice worldwide [61,62,63]. In addition, extensive evidence shows that the words used concerning diabetes affect the physical and emotional health and wellbeing of people living with diabetes, impacting their engagement in healthcare [49]. Speight et al. suggest that changing the language of diabetes can make a powerful and positive difference in the self-care and health outcomes of people affected by diabetes [49]. The environmental scan identified that person-centred language was a major strategy used for supporting diabetes health literacy. Diabetes Australia and the NDSS promoted alignment with the Diabetes Language Statement at all points.

*IV.* 
*Understanding and Actionability*


The environmental scan identified that improving understandability and actionability was important to Diabetes Australia and the NDSS, and they welcomed expert advice. Findings from the PEMAT evaluation demonstrated that DSME fact sheets improved throughout the agreement periods between Diabetes Australia and NDSS during 2016 and 2021 on five different topics. All achieved adequate understandability and actionability, except the actionability of one fact sheet; this is a significant achievement in contrast to other countries. For example, an American study using PEMAT and assessing health information from diabetes organisations found that most patient education materials reviewed scored poorly, with only one-third achieving over 70% for understandability and only one achieving over 70% and meeting the criteria for actionability [40].

#### 4.1.3. Group Reflexivity

The third facilitator of organisational change identified from this study was *group reflexivity*, i.e., “the extent to which group members overtly reflect upon, and communicate about the group’s objectives, strategies (e.g., decision-making) and processes (e.g., communication), and adapt them to current or anticipated circumstances” [64]. Expert health professional advisory councils and consumer groups provided mechanisms for the deliberate process of elaborating on and analysing information and discussing goals, processes and outcomes related to improving organisational health literacy. Diabetes Australia appeared to support a decision-making process that balanced advocacy and inquiry to promote health literacy. Schippers and Rus suggest that focusing on widening the array of opinions contemplated and group reflexivity produces sounder decisions [64].

This was seen, for example, through the systematic, independent appraisal process developed for written information. MESAC meetings, which included a broad and diverse array of opinions, were used to critically reflect on how information could better meet the intended audiences’ needs in alignment with health literacy demands. Simultaneously engaging people from the diabetes communities affected by the recourse or programme to support meaningful inclusion [65]. Likewise, the NDSS Priority Area Leaders followed a similar process in developing information for people with diabetes. The reflective inquiry identified areas enabling efficiencies and increased reach with the aim of improving health literacy. This reflective process aligns with evidence from Beck et al. that shows self-reflection and the critique of practice aid continual changes and advancements in practice [66].

### 4.2. Future Considerations

Diabetes is Australia’s fastest-growing health condition [8,25,26]. Enhancing health literacy is essential for diabetes self-care and optimal health outcomes; evidence indicates that lower health literacy is associated with increased diabetes, diabetes-related complications and adverse health outcomes [11,12,13,14]. In response to this, Diabetes Australia has taken an organisational approach to address health literacy, implementing systematic changes at the policy and practice levels. Our findings suggest that the process has reduced the health literacy demands that resources place on people with diabetes, which may assist people living with diabetes in understanding their condition better and empowering them to take correct self-management actions to improve their health and wellbeing [14].

Our findings also point to areas of future research and practice. The absence of visual aids, such as graphs, tables, charts and pictures, was identified as a key contributor to the lower understandability and actionability of diabetes information [40]. Using non-textual information such as visual aids is effective in conveying complex health information [19]; therefore, the fact sheets might miss the opportunity to enhance the reading comprehension of people living with diabetes effectively.

Moving forward, increased opportunities for incorporating easily understood language and graphics in diabetes health information co-designed and tested with the target audience and delivering this multimodally (e.g., print, audio–visual and social media content) is a priority. Wayfinding performance testing, which observes and evaluates how people orient themselves in physical or virtual spaces and navigate from place to place, may also support organisational health literacy change and responsiveness [53].

In terms of future research, it may be valuable to undertake a more comprehensive analysis, including the full spectrum of DSME fact sheets. Notably, we did not explicitly find diabetes-specific health literacy practice change addressing young adults. Given that the 18 to 24 age group found it more challenging to engage with health providers and navigate health systems in the last “National Health Survey” [6,43], it would be essential to address this omission to align with current national diabetes strategies [51].

### 4.3. Limitations

There are several limitations of this study. Some information collected from the key informants relied on memory. Although the PEMAT offers a critical evaluation of the quality of fact sheets, it is limited in scope; for example, it needs to assess the cultural suitability and linguistic nuances of translated fact sheets that impact health literacy [67]. The decision to include two researchers for the PEMAT assessment is a strength and a weakness. The researchers had a contextual understanding of diabetes, contributing to their assessments of understandability or actionability. As such, their assessment might be more reflective of people with higher health literacy. Finally, we are not able to establish causal links between the policy and practice changes identified in our environmental scan and the PEMAT results.

## 5. Conclusions

Diabetes Australia, as the National Diabetes Services Scheme administrator, has enacted evolving policies to improve organisational health literacy. In this way, health literacy has become integral to the objectives, structure, and operations of the NDSS. Leveraging national-level policies and data, employing an incremental approach supported by a health literacy guided checklist, and group reflexivity appear to have supported organisational changes, and the health literacy demands of diabetes information has concurrently improved. A streamlined quality information development process at the national level has been formed as a first step towards improving organisational health literacy.

## Figures and Tables

**Figure 1 ijerph-20-05778-f001:**
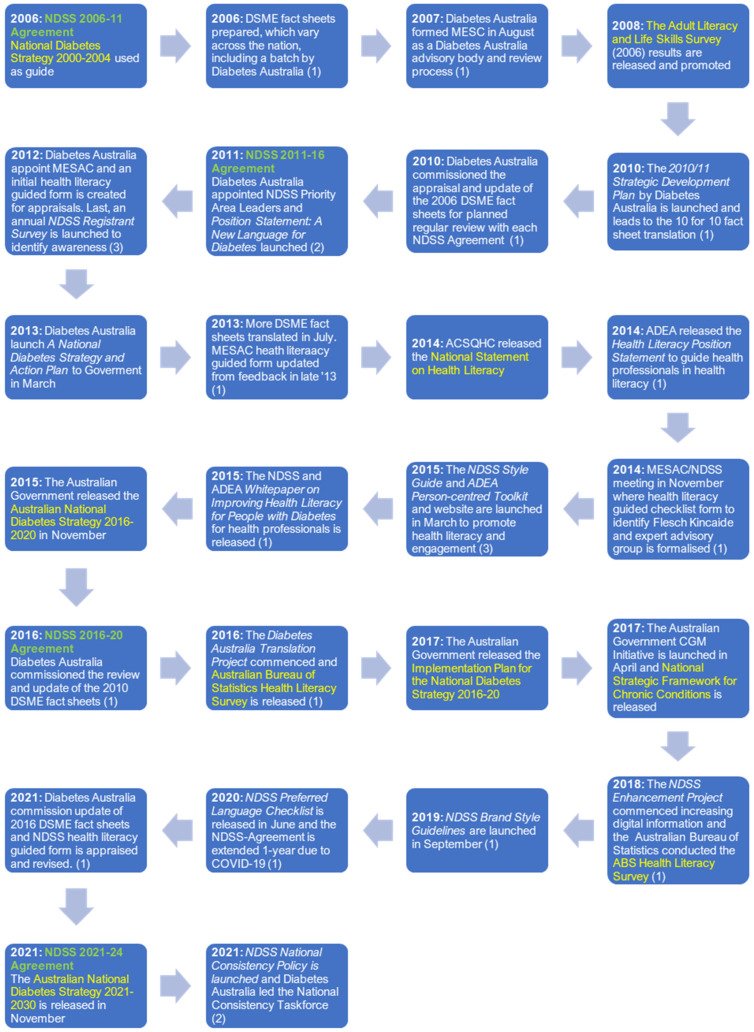
Chronology of NDSS Agreements and national policies/surveys guiding diabetes health literacy practice changes.

**Figure 2 ijerph-20-05778-f002:**
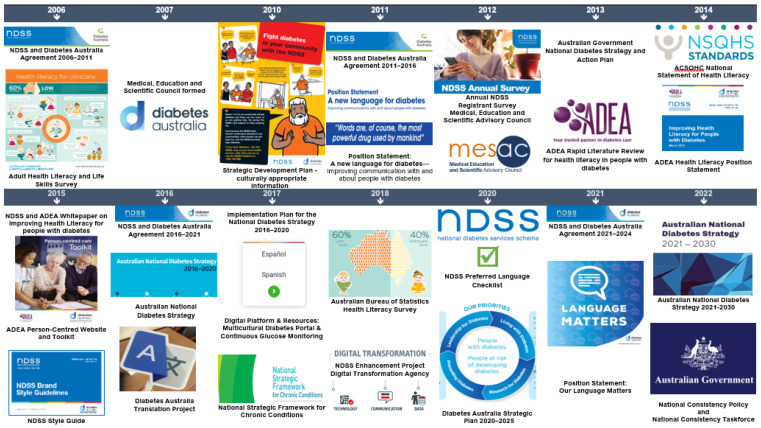
Key diabetes policy and position statements impacting health literacy practice change.

**Figure 3 ijerph-20-05778-f003:**
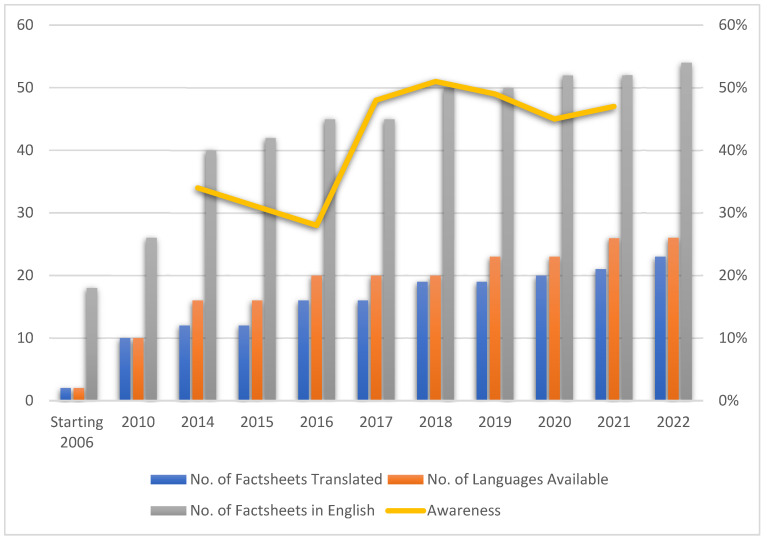
Numbers of diabetes self-care fact sheets accessible to and translated for people living with diabetes and their awareness among NDSS registrants. NB: Some fact sheets’ development and updates fell outside the specific year of review and updates due to the fact of changes in evidence.

**Table 1 ijerph-20-05778-t001:** Mean understandability and actionability from the PEMAT-P assessment by year.

FACT SHEET	2006		2010		2016		2021	
	Under-Standability (%)	Action- Ability (%)	Under- Standability (%)	Action- Ability (%)	Under- Standability (%)	Action- Ability (%)	Under Standability (%)	Action- Ability (%)
Category GENERAL								
Prediabetes	50	33	57	33	63	40	81	67
Category LIFESTYLE								
Coeliac diseases and diabetes	57	67	53	67	63	80	71	100
Glycaemic index	43	17	53	50	94	83	88	83
Category MEDICAL								
Sick days for type 2 diabetes	43	33	47	71	81	100	81	80
Staying well with diabetes/diabetes-related complications	71	67	77	60	75	60	75	80

## Data Availability

The data presented in this study are available upon request from the corresponding author. Most information is publicly available online and accessible.

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
