# Peer review of "Shifts in Diabetes Health Literacy Policy and Practice in Australia—Promoting Organisational Health Literacy"

_ijerph, 2023, doi:10.3390/ijerph20105778_

Round 1
Reviewer 1 Report (Previous Reviewer 3)
1. The title of the study is “Shifts in diabetes health literacy policy and practice in Australia – promoting organizational health literacy”. Is this an appropriate title? From 2006 to 2021, there are various activities to inform diabetes and chronic diseases, and there are printed materials, educational materials, and surveys. And it is difficult to understand why “promoting organizational health literacy” was added as a subtitle. Why organizational literacy?
2. If the analysis was performed with secondary data, it is necessary to clearly indicate in the research method how the data was investigated, with what criteria and on what grounds, and through what process the qualitative analysis results were presented. . After the research method is clarified, the implications of the results can be looked into more closely.
3. Is the Patient Education Materials Assessment Tool (PEMAT) analysis a method in which the researcher evaluates and presents the results immediately? Who rated, and how many? As for the analysis data, you said “Documents were selected based on their content relevance to the inquiry”. Is this being evaluated by the researcher?
Author Response
Please see the attachement.

Reviewer 2 Report (Previous Reviewer 1)

Author Response
Please see the attachment.

This manuscript is a resubmission of an earlier submission. The following is a list of the peer review reports and author responses from that submission.
Round 1
Reviewer 1 Report
Title: Shifts in diabetes health literacy policy and practice in Australia – promoting organizational health literacy.
Reviewer Comments:
Diabetes Australia, as the National Diabetes Services Scheme administrator is implementing new policies to improve organizational health literacy. By leveraging several national level policies, incremental approach, and group reflexivity have improved the health literacy demands of diabetes data. A streamlined quality information development process at the national level is the preliminary step towards improving organizational health literacy in Australia. Scheme administrator used expert advisory as a mechanism to support organizational health literacy. This in turn was supported by a systematic process.
Strengths:
1. One of the strengths of this study is the aims of the article.
2. Every year the number of facts sheets are translated into many languages. This is directly proportional to the awareness.
3. Mean understandability and actionability from the PEMAT-P increased from 2006-2021.
Weaknesses:
1. Quality of the figure 2 can be improved. Some of the data is not visible.
2. Language barriers and reluctance from the people to get updated information will be an issue.
3. Most of the data used for the analysis was obtained from third parties.
4. Figure 1 can be improved.
Author Response
Thank you for this comments.

Reviewer 2 Report
Title: Avoid full stop at the end of the title of the manuscript
Abstract:
1. Authors need to describe the websites where they undertook an environment scan
2. Line 32 should read between 2006 and 2021
3. Lines 33-35, what do the percentages in the phrase in the bracket, 79% and 82%, respectively, refer to?
4. What is the implication of undertaking this study that could be described in the abstract? Is it a kind of audit study, then?
Introduction
1. Lines 45-47, in the statement, while international estimates vary, systematic reviews and large-scale surveys have identified that between 47% and 55% of adults have low health literacy [3,4], costing national governments at least $106 billion annually. Are the figures global data or Australian Data?
2. Lines 47-49: the statement requires further explanation. What do the authors mean by ‘adequate’ health literacy? What do the authors mean by significant social disparities, and would they provide figures about the disparities by age, language spoken at home, and rurality?
3. Line 66, ….To bridge this gap, …which gap?
4. Authors did not provide information about the burden of diabetes in Australia, and how the burden could be ameliorated through health literacy.
Overall, the introduction requires a rework and further explanation of issues related to diabetes health literacy, policy, and practice in Australia
Methods
1. In lines 81-82, the statement that reads , Subsequently, accessible and health literacy friendly health information is vital in this context, is unnecessary detail.
2. I am not sure why ethics approval was not sought fo this study. Expert consultation and panels were undertaken to evaluate PEMA, and how were the particpants recruited and involved in the study without ethics approval?
3. Line 118, a brief description of what was consulted to the experts needs to be provided. During expert consultation, what inputs were received from the experts?
Results
1. In lines 209-2010, authors staed, Until 2007, State and Territory diabetes associations
independently developed all DSME information (i.e. ‘fact sheets’) and programs, leading to duplication [31]. What was the impact of independent DSME information sheets by each territory on health literacy practice nationwide and/or within each state?
Discussion
What key findings can the authors describe in the discussion?
Implications of this study for future practice and policy have not been described.
Author Response
Thank you for this comments.

Reviewer 3 Report
1. One researcher did all the work of analyzing the content and dividing it into major themes for one of the important issues. This is judged to be the biggest problem of this study. Content analysis of any scientific research is never done by a single researcher.
2. Present the entire process of the research in a figure so that readers can see the entire framework of the research at a glance.
3. Please specify in the research method how many researchers evaluated how many prints with the PEMAT tool, and also present the average and standard deviation of the scores.
4. 4. Present Figure 1 as a table. In Figure 1, please organize the picture so that you can understand the entire research process and the main contents of each process. (As mentioned in the second comment above)
Author Response
Thank you for this comments.
